# A multiscale electro-metabolic model of a rat neocortical circuit reveals the impact of ageing on central cortical layers

**Sofia Farina**[ID][1]*, **Alessandro Cattabiani**[1], **Darshan Mandge**[ID][1], **Polina Shichkova**[ID][1,2], **James B. Isbister**[1], **Jean Jacquemier**[1], **James G. King**[1], **Henry Markram**[1,3], **Daniel Keller**[ID][1]

**1** Blue Brain Project, École polytechnique fédérale de Lausanne (EPFL), Geneva, Switzerland, **2** Biognosys AG, Schlieren, Switzerland, **3** Brain Mind Institute, École polytechnique fédérale de Lausanne (EPFL), Lausanne, Switzerland

* sofia.farina@outlook.com (SF)

**Data availability statement:** The code associated with the simulations and figures is

## Abstract

The high energetic demands of the brain arise primarily from neuronal activity. Neurons consume substantial energy to transmit information as electrical signals and maintain their resting membrane potential. These energetic requirements are met by the neuro-glial-vascular (NGV) ensemble, which generates energy in a coupled metabolic process. In ageing, metabolic function becomes impaired, producing less energy and, consequently, the system is unable to sustain the neuronal energetic needs. We propose a multiscale model of electro-metabolic coupling in a reconstructed rat neocortex. This combines an electro-morphologically reconstructed electrophysiological model with a detailed NGV metabolic model. Our results demonstrate that the large-scale model effectively captures electro-metabolic processes at the circuit level, highlighting the importance of heterogeneity within the circuit, where energetic demands vary according to neuronal characteristics. Finally, in metabolic ageing, our model indicates that the middle cortical layers are particularly vulnerable to energy impairment.

## Author summary

The brain's remarkable ability to process information comes at a high energy cost. Neurons, the brain's communication cells, require substantial energy to send electrical signals to one another and to remain prepared for communication. This energy is provided through a complex partnership between neurons, glial cells, and blood vessels. However, as we age, this system becomes less efficient and struggles to meet the brain's energy demands. To investigate these processes, we developed a computational model that integrates the electrical activity of neurons with the metabolic functions of surrounding cells and blood vessels. Using a detailed reconstruction of a rat neocortical

available in the following GitLAB repository: https://github.com/BlueBrain/Molsys-MetabolicModel/tree/main/Multiscale_electrometabolic_ratneocortex. The data can be found in 10.5281/zenodo.14187063.

**Funding:** All authors in this study were supported by funding from the Blue Brain Project, a research center of the École Polytechnique Fédérale de Lausanne, from the Swiss government's ETH Board of the Swiss Federal Institutes of Technology. The funders had no role in study design, data collection and analysis, decision to publish, or preparation of the manuscript.

**Competing interests:** The authors have declared that no competing interests exist.

microcircuit, we examined how energy production and consumption are linked to electrical signalling. Our findings reveal that neurons with different electrical properties have distinct energy requirements, and certain layers of the brain are more susceptible to energy deficits with ageing. This research offers a comprehensive framework for understanding the interplay between electrical activity and metabolism and provides valuable insights into how the brain's energy supply is affected by ageing, which may guide future studies on age-related cognitive decline or neurological disorders.

## Introduction

Electro-metabolic coupling, or the relationship between neuronal signalling activity and the metabolic processes that supply energy to sustain it, is fundamental to brain function. Despite its small size, the energetic demands of the brain are remarkably high, accounting for 20% of the body's energy production [1–4]. This energy, in the form of adenosine triphosphate (ATP), is used mainly to maintain ionic gradients across neuronal membranes and is essential to generate action potentials and support synaptic transmission. Thus, action potentials and postsynaptic potentials are the main consumers of ATP, followed by the regulation of the neuronal resting potential [4]. In neurons, the restoration of ionic gradients relies on the sodium–potassium pump, also known as $Na^+$-$K^+$ pump, which is the most energy-demanding molecular mechanism.

Brain energy production relies on a complex interplay between blood flow, neurons, and astrocytes. This neuro-glia-vasculature (NGV) unit coordinates energy production and distribution, integrating signals from neurons and glial cells to regulate blood flow and deliver metabolic substrates. Blood vessels and capillaries supply the brain with nutrients and oxygen, which astrocytes and neurons then utilize for energy production. Astrocytes play a critical role as metabolic mediators between neurons and blood vessels: they regulate blood flow and convert glucose into lactate, which is then delivered to neurons to fuel ATP production [5–7].

Given the intricate relationships between neurons, astrocytes, and the vascular system, brain energy dynamics are highly sensitive to disruptions in any of these biological processes. Ageing can alter these metabolic processes and is recognized as a significant risk factor for neurodegenerative diseases [8]. It also significantly affects brain metabolism, primarily through a global decrease in cerebral blood flow, reducing oxygen and nutrient delivery while altering glucose metabolism [9]. This decline contributes to neurodegeneration, which is often associated with a reduction in brain volume [10]. The loss of brain volume reflects neuronal atrophy and synaptic loss, processes that are closely related to the lack of energy supply. Different regions of the brain exhibit varying susceptibilities to ageing and neurodegenerative diseases [11], with regions rich in synapses and possessing long-range axons being particularly vulnerable [12]. Many aspects of electro-metabolic coupling remain poorly understood, especially how the energy demands of neuronal activity are integrated and regulated at the circuit level by different cell types. The energy requirements of neurons are not uniform and can vary according to cell type, firing patterns, and synaptic activity.

Computational models have been proposed to investigate electro-metabolic coupling [13–15] by considering a unitary NGV ensemble. These models are often compartmentalized and described through systems of ordinary differential equations (ODEs). The advent of supercomputers and the ability to reconstruct neuronal circuits *in silico* [16] have enabled large-scale simulations that facilitate the study of complete circuit behavior, including neuronal morphologies, heterogeneity, and synaptic interactions [17–20]. Although metabolically coupled ODE models provide valuable insight at the single-unit level, they are constrained by

assumptions of homogeneity, overlooking detailed morphologies, electrical types, and neuronal interactions. Circuit models, on the other hand, have achieved high levels of detail in neuronal electrophysiology, but primarily focus on neurotransmission and do not account for energy supply mechanisms [20,21].

To bridge the gap between electrophysiology and metabolism, we develop a multiscale framework of a reconstructed rat brain neocortex that integrates an electrophysiological model with a metabolic model of NGV across multiple scales (Fig 1A). The circuit design is derived mainly from anatomical reconstruction [17] and an electrophysiological framework [18], both of which are based on previous foundational research [16]. The anatomical model provides a detailed reconstruction of the neuronal circuit, while the electrophysiological component incorporates detailed neuronal electrical properties [19]. The electrophysiological model is combined with a comprehensive metabolic model [15], as schematized in Fig 1B, which describes the metabolic supply chain of NGV through ODEs.

This multiscale model provides a state-of-the-art framework for exploring how dynamic energy management supports and regulates neural activity and how it is altered in different states and diseases. After describing and validating our framework, we demonstrated its application by characterizing the energetic cost of electro-metabolic coupling and examining how electrical features varied across layers and neuron types within the circuit. The extensive network enabled a statistical analysis of the impact of layers and cell types on electrical properties. We also simulated age-related changes in metabolic rates to explore the effects of metabolic ageing, without accounting for structural or cellular density changes, enhancing the framework's ability to model neocortical dysfunction. Fully open source, it offers high detail and adaptability, allowing applications at the scale of entire reconstructed neocortical microcircuits or smaller neuronal samples. Furthermore, it integrates seamlessly with other metabolic models represented as ODEs, providing a foundation for advancing research on electro-metabolic coupling.

## Results

### Model overview

We implemented a reconstructed rat neocortical circuit (Fig 1A) following the approach described in the literature [16,17]. The circuit consists of neurons, astrocytes, and vasculature elements [19,22,23]. Neurons were morphologically reconstructed [24,25] and distributed according to their natural densities [26] accounting for realistic layer-specific proportions and the balance of inhibitory and excitatory cells. The reconstructed circuit contains 129,348 neurons and is described in the section herein entitled Reconstructing a rat neocortical microcircuit. For this work, we extrapolated a microcircuit maintaining the original proportion of the layer comprising 27,962 neurons, which occupies a volume of $400 \times 600 \times 1400 \ \mu\text{m}^3$.

The integration of neuronal electrophysiology with NGV metabolism is based on the exchange of ATP between the two systems, as illustrated in Fig 1B. The metabolic model generates energy through complex pathways and mitochondrial activity, while the electrophysiological model consumes it via $Na^+ - K^+$ pump activity, modelled according to previous work [27]. A crucial aspect is properly addressing the different timescales of the two biological processes: electrophysiology operates on the millisecond scale, whereas metabolic processes occur over a much longer timescale, ranging from seconds to minutes. To account for this, we introduce a coupling time during which the two systems exchange ATP information. Fig 1C illustrates that, from the start to the end of a simulation, multiple coupling times occur. Between consecutive coupling times, both the electrophysiological and corresponding metabolic NGV components for each neuron in the circuit are solved

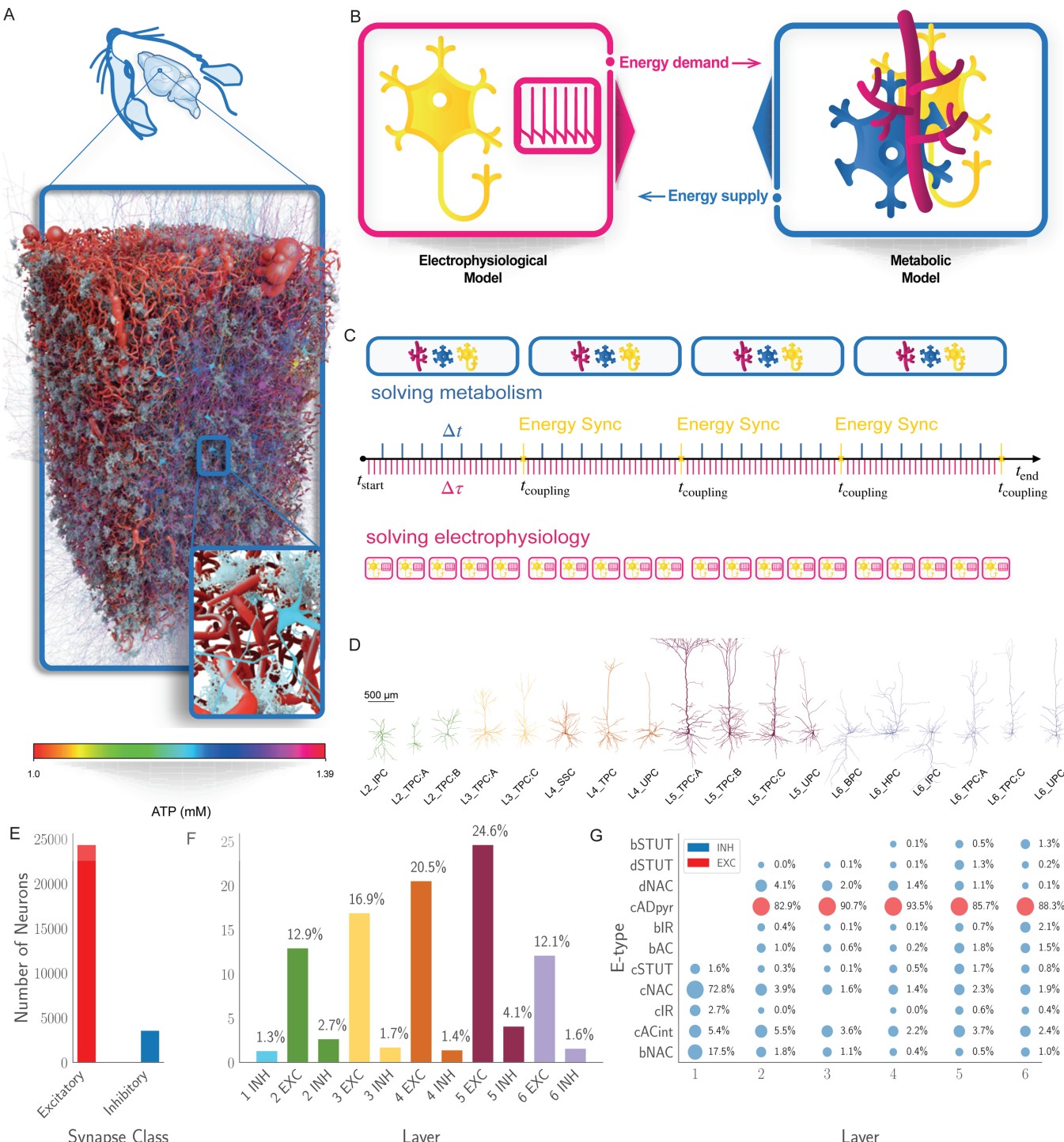

**Fig 1. Model description and profiling the reconstructed rat neocortical circuit.** A: Reconstructed visualization of the reconstructed rat neocortical microcircuit with vasculature, astrocytes, and colour-coded neurons based on their energetic level (ATP) obtained by our simulation. Note that while the vasculature and astrocytes are shown for visualization purposes, their morphological and geometrical data are present in the microcircuit but are not used in the simulation. B: Schematic overview of the coupling between the electrophysiological and metabolic models. C: Illustration of the time-coupling between electrophysiological and metabolic models to address the different timescales. Electrophysiology operates on a much faster time scale compared to metabolism. As a result, the two processes run in parallel with different numerical time steps and are synchronized at coupling intervals based on the slowest metabolic timescale. D: Morphological reconstruction of neurons in the circuit [18], with each neuron colour coded according to its respective cortical layer. E: Histogram showing the number of excitatory neurons (EXC) versus inhibitory neurons (INH) in the neocortical microcircuit. F: Layer and synaptic composition as percentage of the microcircuit. G: Microcircuit composition of electrical types (e-types) per layer in logarithmic scaling to sizes based on percentage.

simultaneously and independently. Specifically, the electrophysiological model was solved using NEURON [28,29], with a fine temporal resolution, while the NGV metabolic system was solved in Julia using the Rosenbrock23 solver over the same period. At each $t_{coupling}$, ATP concentrations were synchronized to ensure consistency between the energy produced by metabolism and the energy consumed by electrophysiology, while ADP levels were updated accordingly. In our simulations, the coupling time was set to every 100 ms to balance the need for sufficient energetic production while preventing complete energetic depletion, ensuring biological realism. For computational efficiency, neurons were distributed across ranks, and the entire computation was parallelized (further details are provided in Section Materials and methods).

Neurons have different electrical (e-types), made with parameter-optimized models [19] and morphological types (m-types), as illustrated in Fig 1D (see Electrophysiological model and Table A in S1 Text). Each single cell was integrated into Neurodamus [30], a software framework that simulates the electrophysiological activity of the entire circuit. Compared to the original approach [18,19], this version includes adaptations for interaction with metabolism via ion-specific mechanisms. Tables B and C in S1 Text describe the channels, pumps, co-transporters and ion dynamics mechanisms added to different e-types.

The metabolic model, consisting of a system of ODEs, represents metabolic processes within the NGV assembly [15]. It encompasses several cellular compartments, including the neuronal and astrocytic cytosol, mitochondrial matrix and intermembrane space, interstitium, basal lamina, endothelium, capillaries, arteries (with fixed arterial concentrations of nutrients and oxygen), and the endoplasmic reticulum (with a fixed $Ca^{2+}$ pool). Enzymes and transporters define the rate equations that govern the dynamics of metabolite concentrations. S1 Fig shows a schematic overview.

Although astrocytic morphologies [22] are available in the microcircuit, we do not explicitly incorporate them into the mathematical framework. However, their volume fractions were included in the ODE system [14].Additionally, layer-specific mitochondrial densities [31] were also included to capture metabolic heterogeneity between different cortical layers. Neuronal heterogeneity was addressed by considering variations in the m-types and e-types. Layer-specificity is further reflected in the circuit through the e-model. For example, pyramidal models for layers 2/3, 4, 5 and 6 were optimized for electrical features specific to their respective layers. We built me-types for various layers with reconstructed or cloned morphologies from corresponding layers, ensuring that layer-specific electrical and morphological properties enhance the biological realism of the circuit. Fig 1E illustrates the total number of excitatory (EXC) glutamatergic pyramidal cells and inhibitory (INH) GABAergic cells within the microcircuit, highlighting the balance between excitatory and inhibitory populations. Neocortical layers span from the top of layer 1 to the bottom of layer 6, and Fig 1F presents the composition of the microcircuit layer, highlighting the distribution of EXC and INH neurons between layers. Additionally, Fig 1G details the distribution of e-types between layers, specifying their presence and proportions in each layer.

## Model validation

Individual components from single-cell dynamics to circuit-level interactions have been built with a bottom-up approach and have been validated in previous studies [15,17–19]. Each e-model was extensively optimized and validated using multiple stimulus protocols, ensuring robustness across different conditions before being generalized to the full circuit [19]. Similarly, the metabolic model was thoroughly tested through sensitivity analyses and diverse stimulation protocols to confirm its stability and physiological accuracy before

integration [15]. Given these previous validations, we aimed to validate the coupled framework, investigating whether the combined system simulated biologically plausible concentrations and the expected circuit behaviour. In particular, we were interested in the balance of energy demand and response.

We ran two main simulations on the microcircuit for 3000 ms: one in which neuronal activity operated with a constant energy level (ATP = 1.38 mM [15]) without activating the metabolic processes, and another in which both the electrophysiological and metabolic components were active, exchanging information every 100 ms. The length of the simulation duration was chosen to capture transient metabolic dynamics and neuronal firing, while balancing computational efficiency. To obtain *in vivo* firing rates, we calibrated layer-specific input stimuli using a relative Ornstein–Uhlenbeck type of stimulus [18]. Fig 2A shows the firing rates of inhibitory and excitatory neurons in the simulation with metabolism. These results aligned well with previously reported values in the literature [32]. Furthermore, we validated the multiscale setup by ensuring that the average key concentrations per neuron (ATP, ADP, $Na^+$, $K^+$, $Ca^{2+}$, and Cl) at the end of the simulation ($T = 3000$ ms) remained within the known physiological ranges [13,14,33–36] (Fig 2B). As expected, the simulations confirmed that all the concentration-related variables were in the range of biologically plausible values, indicating that, by combining the two models, the circuit behaved as expected.

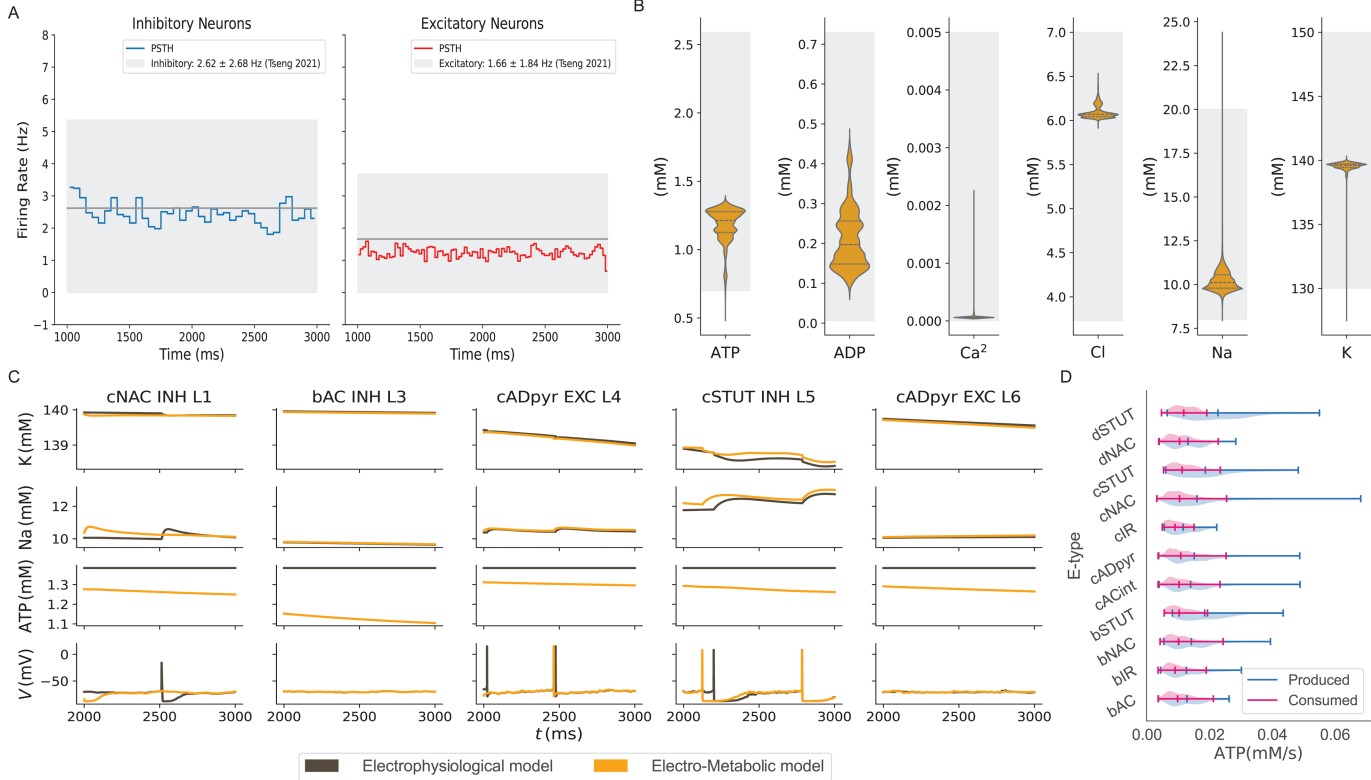

**Fig 2. Model validation and insights on dynamic behaviour in electro-metabolic simulations.** A: The electro-metabolic simulation reproduces the firing rates of inhibitory (blue) and excitatory (red) neurons within the expected physiological range [32] (light grey area). B: At the end of the simulation ($T = 3000$, ms), the average intracellular concentrations of ATP, ADP, $Ca^{2+}$, $Cl^-$, $Na^+$, and $K^+$ in all neurons in the circuit fall within physiologically acceptable ranges derived from various literature sources [13,14,33–36] (light grey area). C: The dynamic behaviour of intracellular $K^+$, $Na^+$, ATP, and voltage for five neurons from different layers and e-types in the electro-metabolic simulation (orange) is compared to the electrophysiological model, where ATP is held constant at 1.38, mM (brown). D: The violin plots compare the average ATP produced (pink) to the average ATP consumed (light blue) at the final simulation time ($T = 3000$ ms) in all neurons, showing that metabolism generates enough energy to meet neuronal demands.

We compared the dynamic behaviour of key variables in five neurons taken from different layers and e-types for the two simulations presented (Fig 2C). The dynamic energy state inside neurons significantly influenced their electrophysiological behavior. ATP was generally lower leading to higher intracellular $Na^+$ and $K^+$, as reduced energy availability limits the rate of the $Na^+ - K^+$ pump. Lower energy levels also affected the action potentials of neurons, as we observed that the same cells might or might not spike at different times, depending on whether only electrophysiological processes were considered or metabolism was included.

Since ATP demand and supply coupled the electrophysiological and metabolic processes, it was important to ensure a balance between ATP production by the NGV unit and consumption by the $Na^+ - K^+$ pump. Fig 2D compares the energy production from metabolism to neuronal consumption in the simulation with active metabolism, measured at the final time point ($T$ = 3000 ms) for all neurons in the microcircuit. The results showed that energy production exceeds consumption, demonstrating that the NGV unit sustained the energetic demands of neurons. This supported the physiological plausibility of the integrated framework.

The violin plot further highlighted variability in energetic demands across e-types. For instance, the dSTUT e-type exhibited high energy demands, consuming and producing more ATP than other e-types. This suggested that dSTUT cells were more metabolically active, possibly due to a higher spiking frequency, requiring increased energy production to support their activity. This aligned with their delayed stuttering firing patterns, which are energetically intensive.

In contrast, the cIR e-type produced and consumed the least ATP, indicating more energy efficient processes or a less active cell. This could explain its continuous irregular firing pattern, which might involve less energy-demanding neural functions.

Finally, the excitatory cADpyr e-type stood out as the most significant contributor to total ATP consumption, consistent with the fact that excitatory cells were the most abundant type in the circuit [37,38]. Additional details are provided in Fig A and Table A of S3 Text.

The combination of the two models produced biologically meaningful simulations, yielding expected firing rates, physiologically plausible concentration ranges, and a dynamic representation of energy derived from a comprehensive metabolic model. The addition of ion-specific mechanisms enabled more realistic interaction between metabolism and electrophysiology, enhancing the biological accuracy of the simulations.

## Layer and electrical characterisation of energetic consumption

Integrating electrophysiology with metabolism allowed investigation of the interaction between energy use and spiking activity. Building on the simulation of the previous section, where metabolism was active, we extracted key features from each of the 27,962 neurons in the microcircuit and performed statistical analyses to characterize the layers and e-types.

Specifically, we investigated the relationship between average energy production and consumption throughout the simulation period by plotting a colour-coded scatter plot based on spike counts (Fig 3A), where cells are grouped by e-type. The results showed a high correlation between energy supply and demand, with correlations greater than 0.95 for each e-type (see Table B in S3 Text). It was interesting to note that cells with high total spike counts (more than 10) responded by producing more ATP. In particular, cNAC, dSTUT, and cSTUT expressed this characteristic firing pattern according to previous work [19] and the observations made in Fig 2D. Positive Spearman correlations were also observed between final ATP production and spike count (Spearman correlations: cNAC = 0.578714; cSTUT = 0.390978;

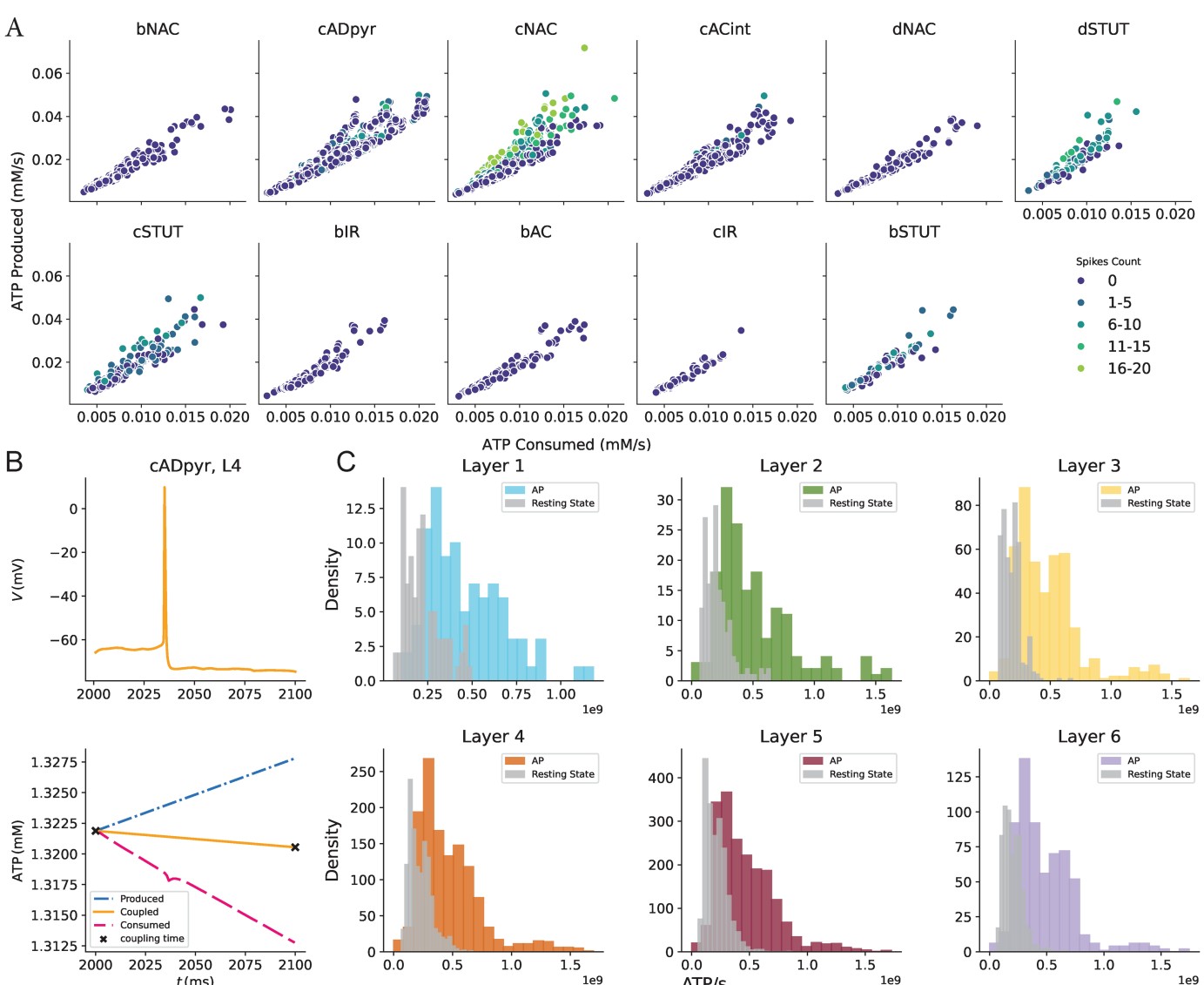

**Fig 3. Correlation between ATP production and consumption across neuronal e-types.** A: Scatter plot of the average ATP produced by metabolism versus ATP consumed by each neuron in the circuit, categorized by e-type. The data points are colour-coded based on the binned number of spikes throughout the entire simulation. B: Dynamical behaviour of voltage and ATP during an action potential (AP) for an excitatory (cADpyr e-type) in layer 4. ATP produced by metabolism (blue), ATP consumed by the electrophysiological model through the $Na^+ - K^+$ pump (fuchsia) and the ATP coupling between the two models (orange). Black crosses highlight the times when coupling between metabolism and electrophysiology occurs (every 100 ms). C: Layer-wise consumption of ATP molecules per AP (colour-coded) compared to consumption during the neuronal resting state (grey). The analysis reveals significantly higher consumption and variability of ATP during activity compared to rest, highlighting the distinct energy demands across the cortical layers.

dSTUT = 0.491834), strengthening the link between metabolic activity and neuronal spiking activity.

As mentioned above, the $Na^+ - K^+$ pump was essential for electro-metabolic coupling in our framework. It expels $Na^+$ ions from the neuron, consuming ATP in the process. Fig 3B shows the dynamic response of ATP, to an action potential (AP): as $Na^+$ entered the cell and $K^+$ was depleted, the voltage-dependent $Na^+ - K^+$ pump ramped up ATP consumption to expel $Na^+$. Since this process is the most energy intensive, analysing ATP depletion in the

electrophysiological system illustrated its effect on overall energy consumption. In Fig 3C, we compared the number of ATP molecules consumed per action potential with that consumed during the neuronal resting state. As expected, neurons exhibited substantial consumption of ATP at rest, on the order of $10^8$ mMs$^{-1}$, with consumption increasing nearly tenfold during an action potential, consistent with previous findings [4,39–41]. This analysis revealed significant energy dynamics, summarized in a statistical table presented in Table C in S3 Text. ATP levels were higher during activity in all layers, with layer 2 showing the highest variability and layer 3 showing the lowest levels of ATP and variability at rest. Furthermore, consumption of ATP during an AP was more variable, with a higher standard deviation and an interquartile range compared to the resting state. Statistical tests confirmed that these differences are highly significant (p <0.001 for all layers), highlighting the distinct energy demands and regulatory mechanisms between active and resting states. These results suggested functional specialization of the cortical layers.

In addition, we investigated several electrophysiological features. For each neuron in the circuit, we recorded firing activity and categorize neurons by layer and e-type. We then extracted the number of spikes, the maximum voltage, the AP amplitude, and the resting state potential at the end of the simulation using the feature extraction library [42]. Descriptive tables for these characteristics are provided in Tables D and E in S3 Text and a visual representation of these electrical features is shown in Fig 4. To assess statistical significance, we first confirmed non-normal distribution in our samples using the Shapiro-Wilk test [43], then applied Dunn's non-parametric test [44] with Bonferroni adjustment [45] for pairwise

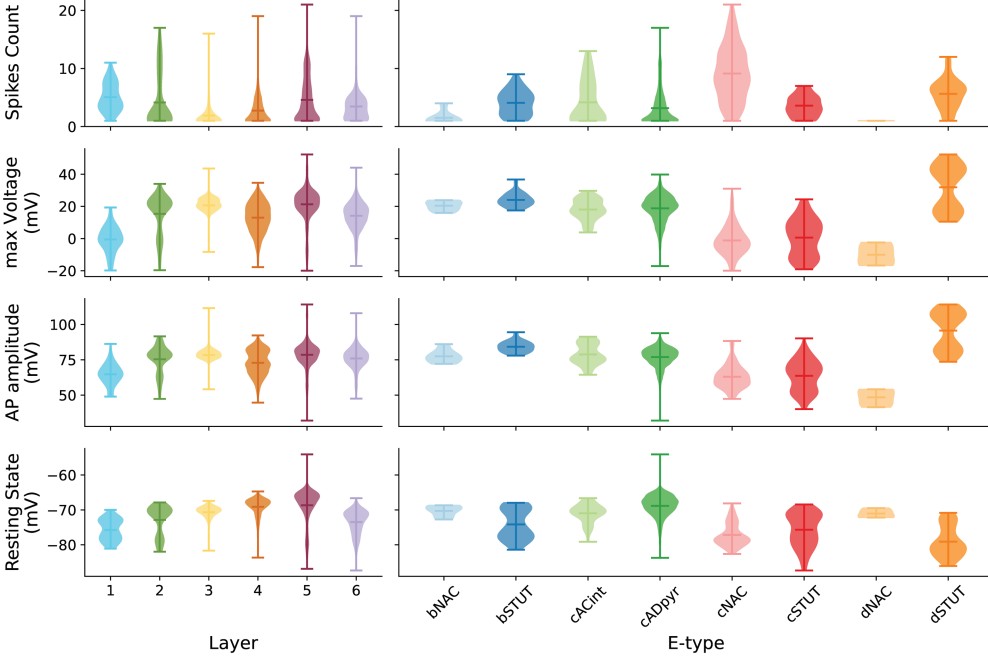

**Fig 4. Layer and e-type characterization of electrical features for spiking neurons.** Violin plots illustrating the distribution of total spike count, maximum voltage, AP amplitude, and resting membrane potential of neurons that fired during the electro-metabolic simulation. Data are grouped by cortical layers (left) and neuronal e-types (right), highlighting significant variations in firing activity, voltage profiles, and excitability across layers and cell types. The plot emphasizes the distinctive electrical properties of specific layers, such as layer 1 having high spiking activity and e-types such as cNAC, bSTUT and cADpyr cells.

comparisons across layers and e-types (see Fig B in S3 Text). This analysis revealed significant electrophysiological differences across layers, reflecting layer-specific variations in neuronal behavior.

Spike count varied significantly between layers, suggesting non-uniform firing rates across the cortex, possibly due to intrinsic neuronal properties or layer interactions. Layer 1 exhibited high spiking activity, primarily due to a high proportion of cNAC cells, averaging 9.5 spikes per neuron. cNAC cells had different firing behaviors, and dSTUT and cADpyr also showed unique patterns in pairwise comparisons (Fig B in S3 Text).

For maximum voltage, significant differences were found between layers, while layers 3 and 5, as well as layers 4 and 6, exhibited similar profiles. bSTUT and cSTUT cells had the highest maximum voltage, followed by bNAC cells, while dNAC cells had the lowest. cADpyr cells showed the widest range of both maximum voltage and AP amplitude. Significant differences in AP amplitude were observed between layer 1 and the other layers, suggesting layer-specific variations in excitability or synaptic input.

For the resting state, significant differences were also found across layers, with layer 5 exhibiting the widest range of values, and layer 1 demonstrating a marked difference compared to the other layers. Pairwise comparisons of e-types revealed notable differences, particularly between bSTUT and cADpyr cells. These findings were aligned with those of [46] with distributions for L5 and L2/3 similar to our results.

Morphologies are an important characteristic included in our electrophysiological framework, accounting for different m-types, as shown in Fig 1D. Therefore, our circuit model allowed for an investigation of the relationship between the neuronal surface area and electrical features we have previously analysed, highlighting the crucial role of including morphologies in computational models. Fig 5A displays a scatter plot for each layer and feature, with the surface area colour-coded by e-types, revealing distinct clusters for certain e-types. Pearson's correlations between the surface area and the four electrical characteristics of neurons with at least 20 samples are shown in Fig 5B.

The correlation between surface area and various features revealed both positive and negative associations across different layers and e-types. Regarding AP amplitude, most correlations were negative, with the strongest negative correlations observed in cADpyr (−0.66) and bSTUT (−0.64), suggesting that as the area increased, the AP amplitude tended to decrease. In contrast, the resting state of the neuron showed mainly positive correlations, especially in cADpyr (0.46) and cNAC (0.32), indicating a trend towards an increase in steady-state voltage with larger area. Regarding spike count, correlations varied, with cADpyr showing a positive correlation (0.44) and cSTUT showing a negative correlation (−0.57), reflecting different behaviours in different cell types. For maximum voltage, stronger negative correlations were observed in cADpyr (−0.62) and bSTUT (−0.68), suggesting that larger areas were associated with lower maximum voltage. These findings highlight how the relationship between area and electrical features varies between different layers and cell types, indicating potential layer- and cell-specific influences on neural circuit behaviour.

In summary, our findings revealed that the cortical layers had different electrophysiological profiles, with more pronounced differences in certain layers. These variations probably reflected functional differences, including excitability, synaptic connectivity, and membrane properties. Accurately representing ATP dynamics in the model revealed the distinct ATP requirements between types of neurons. Moreover, investigating the relationship between neuronal area and electrical properties highlights the need for a more comprehensive approach in future studies, one that integrates both morphology and electrical features into a holistic modeling framework.

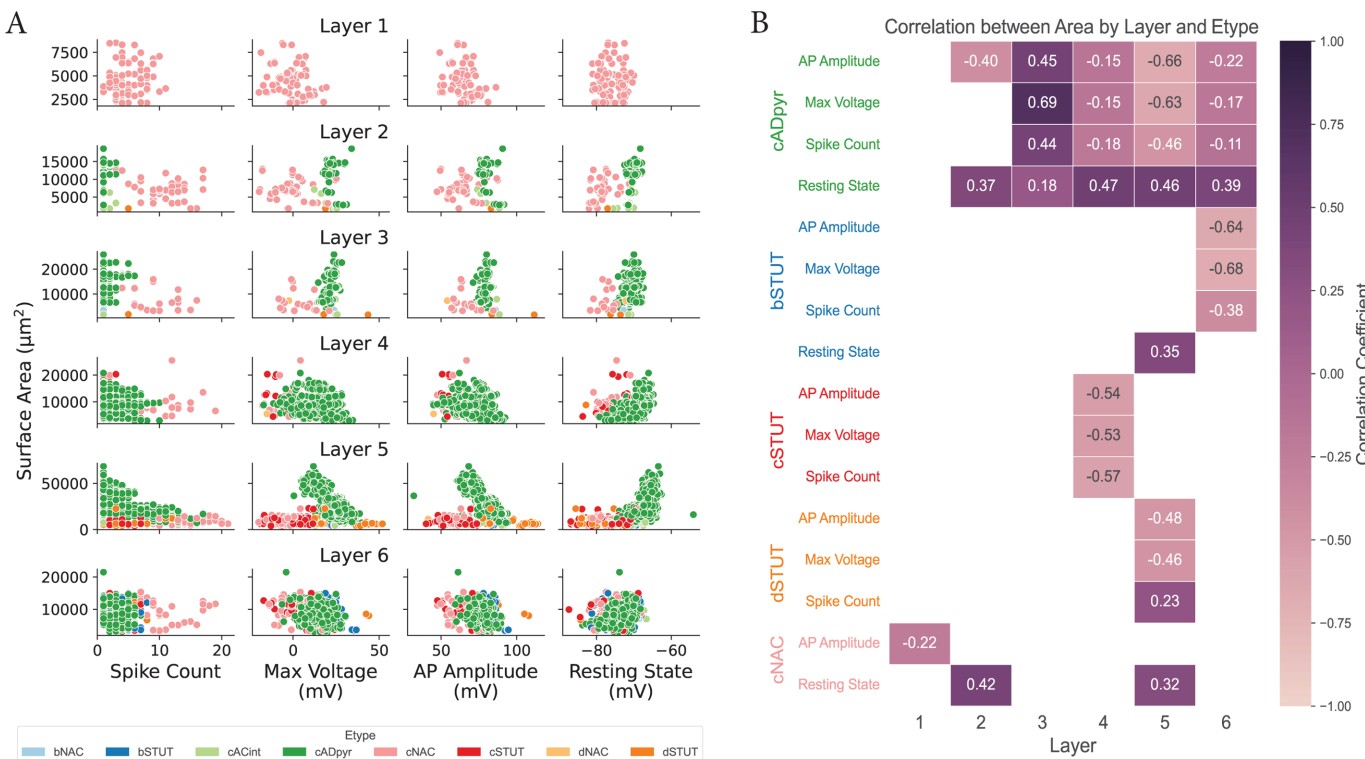

**Fig 5. Clustering of neuronal e-types by surface area and electrical features across layers** A: Scatter plot of neuronal surface area versus spike count, maximum voltage, AP amplitude, and resting membrane potential, grouped by cortical layer and colour-coded by e-type. Different clusters emerge according to the e-type indicating a possible correlation. B: Pearson correlation between surface area and each of the four electrical features (AP amplitude, resting membrane potential, spike count, and maximum voltage) across the cortical layers for each e-type with at least 20 samples and a statistically significant p-value. The results reveal distinct patterns, with negative correlations dominating AP amplitude and maximum voltage, while the resting membrane potential shows predominantly positive correlations, highlighting cell- and layer-specific influences on neural behaviour.

## Effect of metabolic ageing on neuronal electrophysiology

Finally, we explored the impact of ageing on metabolism in the neocortical microcircuit, extending a previous model [15] in which ageing effects were incorporated by modifying enzyme and transporter expression levels according to experimental data from the literature [47,48]. Generally, enzyme expression declines with age in both neurons and astrocytes. However, our model does not account for ageing effects on electrophysiological properties or neuronal morphology.

Building on this earlier work, we simulated ageing in our model by introducing dysfunctions in the mitochondrial electron transport chain (ETC), which affect energy production in both neurons and astrocytes. We adjusted various concentrations of metabolites and cofactors to represent ageing [49,50]. Key changes included a reduction in arterial glucose and intracellular glucose levels [51], downregulation of beta-hydroxybutyrate [52,53], an increase in intracellular lactate [54], and depletion of the total NAD pool (encompassing both NADH and $NAD^+$), reflecting reduced NAD availability with age [55]. Synaptic glutamate release pools were also downregulated to account for ageing [50], while synaptic input was kept constant. Furthermore, the capacity for NADH shuttling between the cytosol and mitochondria was reduced [55].

In this setup, the reference simulation, presented first in Section Model validation, with metabolism maintained in a young state is labelled "Young", while the simulation with applied ageing effects is labelled "Aged". As expected, ageing affected energy production, reducing the available ATP in neurons. Fig 6A illustrates that, per layer, the average total concentration ATP at the end of the simulation ($T$ = 3000 ms) was lower in the aged condition than in the young.

The difference in ATP levels between Young and Aged neurons within the microcircuit is shown in Fig 6B. The scatter plot revealed that ageing appeared to have the greatest impact on layer 4, with an ATP difference of approximately 0.15 mM, followed by layer 3. This suggested that metabolic ageing and its associated energy deficits might originate in the central layers or that these layers might be particularly vulnerable to ageing effects. Their slightly higher mitochondrial volume fraction [31] likely increased their susceptibility to energy production impairments.

As observed, ageing in the metabolic model affected energetic availability in neurons, thus altering neuronal spiking activity. As shown in Table F in S3 Text, which compares the total number of spikes throughout the simulation and the final ATP average for the Young and Aged groups, along with the simulation with inactive metabolism where ATP is fixed at 1.38 mM, our model indicated that greater availability ATP correlated with lower firing rates. This trend could be explained by the dynamic role of ATP concentration in regulating the $Na^+ - K^+$ pump. When this pump is impaired, the firing rates can increase [56], highlighting the critical role of the $Na^+ - K^+$ pump in neuronal function. The decreased availability of ATP resulted in less efficient pump operation, leading to a buildup of intracellular $Na^+$ and a

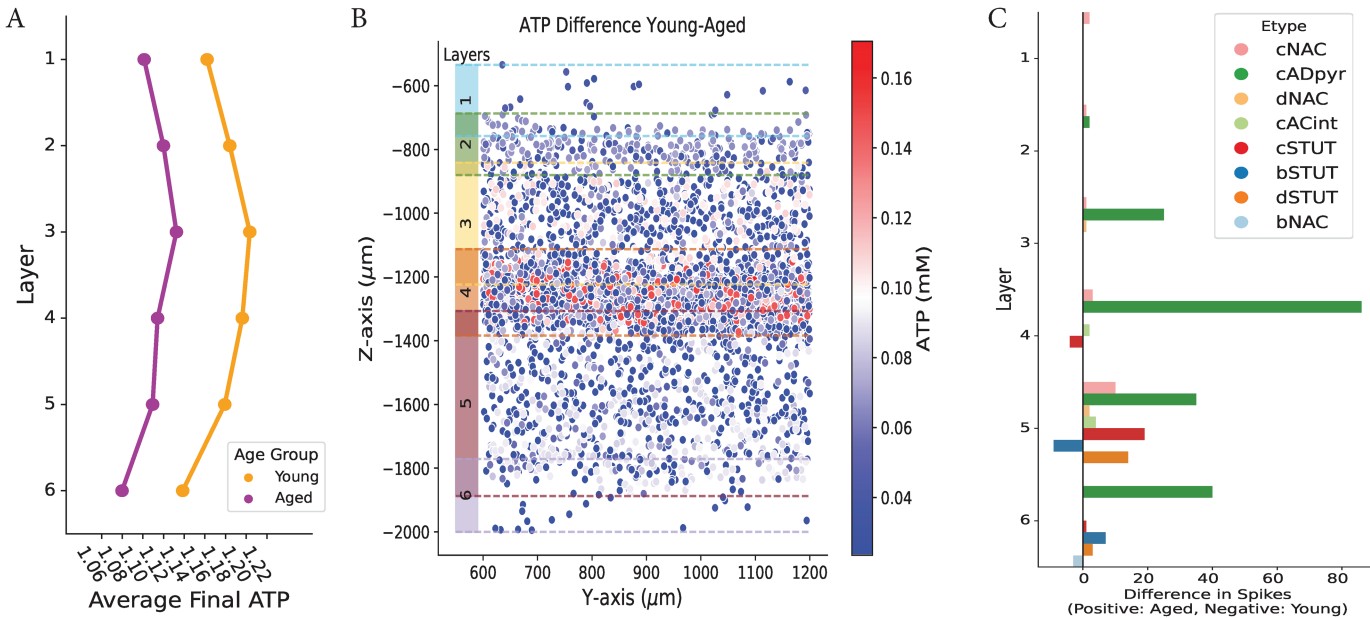

**Fig 6. Effects of ageing on energy and spiking activity in different layers and e-types of neurons.** A: Average neuronal ATP levels in cortical layers at the end of the simulation ($T$ = 3000 ms) for Young (orange) and Aged (magenta) groups. On average, each cortical layer exhibits lower energy levels in Aged compared to Young. B: Visualization of ATP differences between Young and Aged neurons along the $y$- and $z$-axes of the neocortex, highlighting regions with differences greater than 0.1 mM. The left color bar delineates the layers along the $z$-axis, clearly illustrating that layers 3 and 4 are more affected by again. C: Layer-wise histogram of the difference in spiking between Young and Aged clustered by e-types. Positive bars represent spikes that occur in Aged but not in Young, while negative bars indicate spikes that occur in Young but not in Aged.

reduction in $K^+$ gradients. This disturbance in ion homeostasis made it easier for neurons to reach the firing threshold, explaining the increased firing rates observed with reduced ATP.

We further examined differences in spiking activity between Young and Aged neuronal types by plotting changes in spike counts (Fig 6C). Positive bars represent neurons that spike in the Aged condition but not in the Young, while negative bars indicate neurons that spike in Young but not in Aged. In particular, layer 4 exhibited the most pronounced increase in spiking with age, consistent with the previous observation that ATP is more impaired in this layer, leading to higher energetic requirements. Among excitatory pyramidal cells, cADpyr, the largest differences occurred in layer 4, followed by layers 5–6, and then layer 3. Certain inhibitory types, such as cNAC and bSTUT/cSTUT, which are characterised by high spiking frequencies, also exhibited increased activity in the Aged condition.

In general, our findings suggested that central layers might be the first to be affected by ageing, as suggested by [57], who noted that central cortical regions were particularly vulnerable due to their high synaptic density and energetic requirements.

## Discussion

In this work, we presented a reconstructed rat neocortical microcircuit composed of neurons, glial cells, and blood vessels. The electrophysiological model incorporated neurons with morphological and electrical features to capture cortical heterogeneity. Multiple ionic channels were included, with a particular focus on the $Na^+ - K^+$ pump channels [27], which served as a connection mechanism to metabolic processes via ATP consumption. We employed an established metabolic model [15] which represented the NGV unit and solved it for each neuron in the circuit. Different time scales were carefully considered to ensure accurate coupling. Our proposed circuit framework (Fig 1) achieved a high level of biological description by coupling two fundamental processes, making it the first circuit scale model to include such detailed representations, releasing a powerful tool for *in silico* experiments.

The coupled system allowed investigation of neuronal energy requirements and NGV energy supply (Figs 2 and 3). The metabolic component responded by increasing energy production for neurons that had higher energy demands due to elevated spiking activity, highlighting varying energy needs across neuronal e-types. We compared ATP consumption by $Na^+–K^+$ pump during action potentials to that of the resting state. Although consumption of ATP was high to maintain resting potential, it increased further during spiking due to higher demands [4]. Finally, we identified different electrical characteristics linked to cortical layers, surface area and e-types ( Figs 4 and 5). This highlighted the importance of incorporating neuronal heterogeneity to reflect the natural variability of biological processes [58]. Our findings confirmed the specific functions of layer 5 and pyramidal cells [46]. We also characterized the inhibitory cell types and found distinct differences in layer 1 compared to the other layers. The resilience and functionality of biological systems often depend on this heterogeneity, and its loss can affect network performance [59].

Finally, we tested ageing in the metabolic model by tuning various parameters. When comparing the aged simulation to the young one, we observed a clear energetic depletion in the central layers, particularly in layer 4 (Fig 6). This suggested that ageing might first affect the central layers due to their higher mitochondrial density [31]. Regions of the brain with high synaptic density, such as the central cortical areas, are known to be more susceptible to ageing [11], and certain studies highlight a similar vulnerability in the central layers [57]. Furthermore, an increase in glial density and a reduction in neuronal nuclear area in ageing is observed in layers 2–6 [60] suggesting a compensatory glial response to support neurons. This

aligns with our findings, where glial activation in the central layers during ageing may offset metabolic deficits.

Moreover, while our model does not account for dynamic blood flow, we recognize that blood flow plays a crucial role in the maintenance of brain metabolism. The capillary density is particularly high in layers 2–4, with layer 4 being notably dense [61,62]. Although the mechanisms underlying layer-specific blood flow regulation remain debated [63], our findings suggest that ageing may increase the need for targeted blood flow regulation to support metabolic demands, especially in the central layers. If central layers are in fact more vulnerable to ageing effects, their dense capillary networks can play a key role in delivering nutrients and mitigating these age-related deficits. For example, in vivo imaging has shown a significant increase in mural cell density in layer 4 of aged mice compared to young mice, highlighting age-related changes in the cerebrovascular network [64].

Lastly, we demonstrated that firing rates are influenced by energy levels. The $Na^+$ – $K^+$ pump activity was constrained by ATP availability in both the Young and Aged models, and firing rates increase when ATP was limited, similar to the effects observed when $Na^+$ – $K^+$ pump activity is blocked [56].

A model always comes with limitations and assumptions, and although our model captures the key components of both electrophysiology and metabolism, some limitations remain. Currently, the model uses neuronal morphology to solve electrophysiological components, but the metabolic model is applied uniformly for each neuron. A potential next step would be to solve the metabolic model for each individual neuron and astrocyte in the circuit, rather than applying the NGV unit to each neuron alone, which would allow for a more biologically accurate representation of neuro-glial-vasculature interactions. A further advancement would be to solve the metabolic model in morphological-dependent geometries, incorporating both neuronal and astrocytic morphologies, which are already morphologically described in the circuit [17,22]. Preliminary results presented in this work already demonstrate the importance of incorporating geometrical dependence into the model, linking neuronal surface area with electrical features. In particular, astrocytic morphologies play a critical role as metabolic mediators and may significantly influence metabolism [65].

Additionally, our results from simulating ageing with the metabolic model indicate that ageing impairs metabolism in a layer-specific manner. This suggests the need to enhance our simulations with a more detailed model of blood flow [23,66,67]. Incorporating such a model could allow simulation of capillary changes, such as diameter and volume fluctuations, driven by neuronal and astrocytic energy demands. Additionally, previous work has successfully coupled electro-metabolic and hemodynamic models, tracking blood flow through arteries, capillaries, and veins [68], highlighting the feasibility and relevance of incorporating such an approach into our framework. This would help to assess the role of the density of blood capillaries in ageing. Furthermore, incorporating a model of the extracellular space could influence molecular diffusivity and clearance, potentially leading to further model enhancements. For example, previous models have already shown the importance of extracellular potassium [13].

In our study, we have focused solely on metabolic ageing. However, ageing also affects electrophysiology by reducing the distribution of channel density and by affecting neuronal morphologies [69]. Neuronal morphology undergoes dendritic atrophy [70], leading to a reduction in synaptic connections and overall synaptic activity [71]. In the context of our electro-metabolic model, this decline could change energy demands, alter ionic balance, and impact network dynamics, potentially creating a feedback loop that reinforces metabolic impairments observed in ageing.

From a numerical point of view, it is crucial to carefully select the duration of the coupling time intervals to prevent the system from straying away from biologically realistic dynamics while avoiding extra overhead due to frequent syncing of the simulators. Furthermore, the metabolism-coupled circuit is subject to the inherent limitations present in both the metabolic and electrophysiological models. For a more detailed analysis, we refer the reader to the discussion section of the respective work [15,18,19]. Briefly, the electrophysiological model was optimized mainly using somatic patch-clamp recordings, with limited dendritic and axonal data, while the metabolic component relied on literature-based approximations and did not account for differences factors, such as sex differences and again-related changes in oxygen transport.

In conclusion, we presented a detailed framework for a reconstructed rat neocortical circuit that coupled electrophysiology and metabolism. Our model revealed that ageing might first affect the central cortical layers. Since the model is open source, it can be further developed, applied to study any region of the brain, and used to model *in silico* ageing or other neurodegenerative diseases.

## Materials and methods

### Electrophysiological model

The electrophysiological model was implemented using the NEURON simulation environment [28]. To account for neuronal heterogeneity, we incorporated a diverse range of m-types [16,18,24,72] and e-types [19].

We used BluePyEModel [73] software to construct the e-models. It combined BluePyOpt [74] for the optimization and validation of multi-objective models, and eFEL [42] and BluePyEfe [75] for the extraction of features. The model building process is similar to that described in previous work [19]. Neuron models were constructed for 34 °C. Additional model constants and some initial parameters are mentioned in Table D in S1 Text. The circuit contains eleven e-types, presented in Table 1.

These e-types were paired with various morphology types (m-types) (see Table A in S1 Text) to form different morpho-electric types (me-types). The m-types present in the circuit are as follows:

**Table 1. Neuron e-types and their descriptions.**

| e-type | Description |
|---|---|
| **Excitatory e-type** | |
| cADpyr | Continuous Adapting Pyramidal Cells |
| **Inhibitory e-types** | |
| cAC | Continuous Accommodating |
| bAC | Burst Accommodating |
| cNAC | Continuous Non-Accommodating |
| bNAC | Burst Non-Accommodating |
| dNAC | Delayed Non-Accommodating |
| dSTUT | Delayed Stuttering |
| bIR | Burst Irregular Firing |
| cIR | Continuous Irregular Firing |
| bSTUT | Burst Stuttering |
| cSTUT | Continuous Stuttering |

- Excitatory cell m-types: BPC: Bipolar PC; HPC: Horizontal PC; IPC: Inverted PC; TPC:A: Tufted PC, late bifurcation; TPC:B: Tufted PC, early bifurcation; TPC:C: Tufted PC, small tuft; UPC: Untufted PC; SSC: Spiny Stellate Cell.
- Inhibitory m-types: BP: Bipolar Cell; BTC: Bitufted Cell; CHC: Chandelier Cell; DAC: Descending Axon Cell; DBC: Double Bouquet Cell; HAC: Horizontal Axon Cell; LAC: Large Axon Cell; LBC: Large Basket Cell; MC: Martinotti Cell; NBC: Nest Basket Cell; NGC-DA: Neurogliaform Cell with dense axon; NGC: Neurogliaform Cell; NGC-SA: Neurogliaform Cell with sparse axon; SAC: Small Axon Cell ; SBC: Small Basket Cell

This resulted in 212 distinct morpho-electric (me-type) combinations, derived from experimental data of the rat somatosensory cortex [18].

We first created 40 electrical models (e-models) and then generalised these e-models for multiple morphologies using emodel-generalisation software [72]. This process resulted in a total of 129,348 neuron models being used in the circuit. Generalisation involved applying the final mechanism parameters of these e-models to multiple reconstructed and cloned morphologies based on the possible me-type combinations.

Various mechanisms: ion channels, pumps, co-transporters and ion dynamics mechanisms were used to construct electrical models. These included Sodium ($Na_v$) Channels: Voltage-gated transient Na (NaTg), Persistent Na (Nap_Et2); voltage-gated potassium ($K_v$): transient K (K_Tst), persistent K(K_Pst), $K_v$ type 3.1 (SKv3_1); small-conductance calcium-activated K channel (SK_E2), stochastic $K_v$(StochKv3), D-type $K_v$(KdShu2007); voltage-gated ($Ca_v$): high-voltage-activated Ca (Ca_HVA2), low-voltage-activated (Ca_LVAst); hyperpolarisation-activated cation channel (Ih), sodium-potassium-chloride co-transporters (Na-K-Cl Co-transporter (nakcc); sodium-potassium pump (nakpump); Leak channels and an ion dynamics mechanism (internalions) (Table B in S1 Text). Table C in S1 Text shows the locations where these mechanisms are inserted in different neuronal locations with different e-types. We modified these mechanisms from the original work [19] adding new channels to include changes in ionic concentration. In particular, the metabolism model communicates with electrical models using ATP and ADP concentrations. These models were re-optimised and validated to fit the electrical features [19], and also fit the resting- and steady-state ionic and ATP concentrations for different stimuli.

## Metabolic model

The metabolic model was originally proposed by [15]. It consisted of a detailed system of ODEs describing metabolic interactions within an NGV unit combined with a classic Hodgkin-Huxley neuron model, a schematic overview is shown in S 1 Fig. The equations were derived from various sources in the literature [13,14,76–80]. The model is compartmentalized and included the neuronal and astrocytic cytosol, the mitochondrial matrix and inter-membrane space, the interstitial space, the basal lamina, the endothelium, the capillary, and an artery (with fixed arterial concentrations of nutrients and oxygen). It also incorporated the endoplasmic reticulum, which maintains a fixed $Ca^{2+}$ pool.

In this work, the original model had been predisposed to be coupled with an electrophysiological one through the $Na^+$ – $K^+$ pump. Tables A-H in S2 Text show the 183 equations, Tables I-X in S2 Text describe the fluxes, and Tables Y-AB in S2 Text the initial values. Moreover, we account for metabolic variability across the cortical layers by integrating mitochondrial densities specific to each layer [31]. Blood flow was not dynamically modelled, but we considered a fixed level of 0.0001 ml/min and a fixed blood volume of 0.023 ml/min.

### Reconstructing a rat neocortical microcircuit

We used an NGV circuit consisting of neurons, synthesized astrocytes, and the cerebral vasculature [22,23]. It is based on a detailed reconstruction of the somatosensory cortex of a juvenile rat that algorithmically models the precise anatomy, connectivity, and electrophysiology of neurons [16–18]. The neocortical volume is 14,18 mm$^3$, which contains 129,348 neurons with 206 million synapses, 42,362 astrocytes (for a neuron-to-astrocyte ratio of 3:1 [81]), 19 million glial-glial cell connections, 185 million neuron-glial connections and 73,345 glial-vasculature connections. For the simulation presented in this work, we extracted a microcircuit of 27,962 neurons within a volume of 0.336 mm$^3$, maintaining the proportions of the original layer of the circuit.

### Simulation methods and conditions

The multiscale model consists of three primary elements: Neurodamus, Metabolism, and the Multiscale Orchestrator. Fig 7 schematically illustrates the relationships and main interactions among these elements. The following discussion provides an overview of the main components and their interactions.

**Neurodamus.** Neurodamus [30] is a Python-based application designed to control simulations within the NEURON environment, implemented in C++ [28]. NEURON employs a semi-implicit Euler method for time integration, carefully balancing computational performance, accuracy, and stability.

In Neurodamus, neurons are modelled as structured graphs of segments (Fig 1D), with a precise location in space and connected by synapses. Mechanisms, synapses, and electrical currents are all simulated within the segments. The time integration step, set to 0.025 ms, is

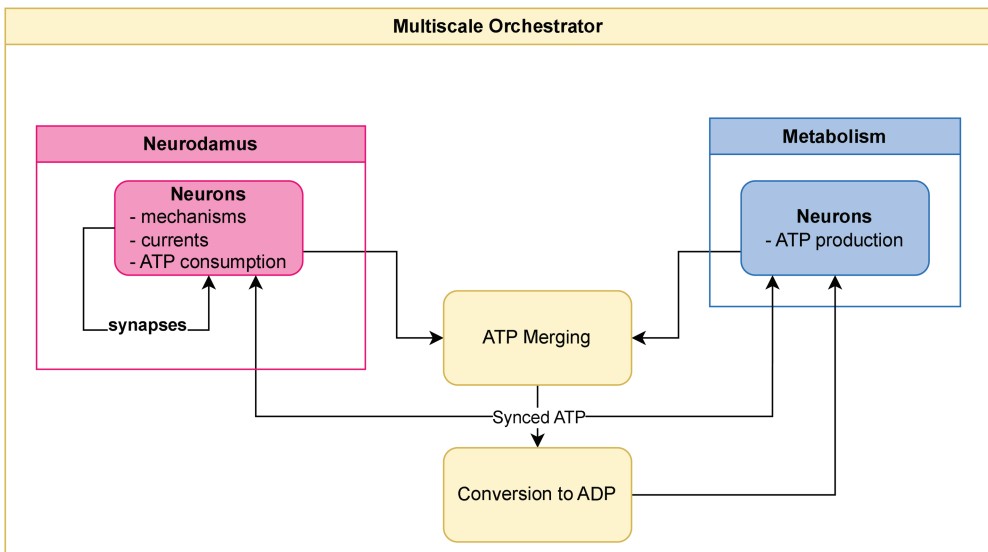

**Fig 7. Key components and interactions in the multiscale model.** Schematic representation of the relationships and key interactions among the components of the Multiscale model: Neurodamus, Metabolism, and the Multiscale Orchestrator. Neurodamus: simulates neuronal activity, covering neuron dynamics, synaptic interactions, ionic currents, and mechanisms that consume ATP. Metabolism: models energetic production within a NGV unit. During synchronization, the Multiscale Orchestrator exchanges critical variables between modules. ATP production and consumption are balanced, after which ADP is calculated and re-injected into the Metabolism simulator.

the finest in the entire model, capturing the intricate dynamics of neuronal activity with high temporal resolution.

The application simulates not only the neurons themselves and their interactions via synapses but also the flow of electrical currents and various cellular mechanisms presented above.

To enhance computational efficiency and scalability, Neurodamus automatically distributes neurons across multiple ranks, allowing for efficient parallelization of the simulation. This approach facilitates the handling of larger and more complex models.

External conditions, such as potassium concentration ($[K]_o$ = 5 mM) and extracellular calcium ($[Ca^{2+}]_o$ = 1.1 mM), are kept constant throughout the simulation, ensuring a stable extracellular environment for neuronal function.

**Metabolism.**   In the Metabolism module, we solve the metabolic model presented in the previous section. Unlike the electrophysiological model, where neuronal morphology is explicitly represented, the metabolic model does not incorporate morphological details, as it is described by a system of ODEs. For each neuron in the circuit, we solve the metabolic model, incorporating the metabolic contributions of neurons, astrocytes, and the vasculature. The primary output of this module is the updated ATP concentration for each neuron.

From the numerical standpoint, the equations governing this system are non-linear, a common characteristic of biological models, leading to a stiff system of equations that presents unique computational challenges. These are managed by solving the system in Julia [82], using the Differential Equations Suite and specifically the Rosenbrock23 solver, which is particularly effective for stiff problems. The time step used in this module is the largest in the overall model, set at 100 ms. This choice is facilitated by the robustness of the Rosenbrock23 solver, which enables integration over a larger time step, thus reducing computational time while maintaining accuracy in the simulation. Additionally, the implementation takes advantage of the neuron distribution provided by Neurodamus and is automatically parallelized to enhance performance and efficiency.

**Multiscale orchestrator.**   The Multiscale Orchestrator, developed in Python, is responsible for coordinating the simulation process across different modules. Its key functions include:

- Initializing the simulators
- Managing the execution order to determine which simulator is active at any given time
- Synchronizing simulators by transferring values, performing area and volume averages, and handling unit conversions
- Recording essential quantities
- Performing sanity checks to ensure that critical quantities remain within specified limits

The orchestration follows a defined sequence to account for the different time scales of the two biological processes, as illustrated in Fig 1C:

1. Neurodamus advances until a synchronization time step.
2. Metabolism then advances to the same synchronization time step.
3. A syncing operation occurs where values are exchanged and converted as needed.

This process repeats at regular intervals. To simplify synchronization, the synchronization interval is set to the smallest common multiple (SCM) of the simulator time steps, which in

this case is 100 ms. Since values are updated during synchronization, discontinuities may appear in concentration and current traces at these specific time steps.

The quantities synchronized between the simulators are:

- neuronal intracellular ATP balanced according to an additive splitting scheme [83], using the following formula:

$$[ATP]_j = [ATP]_{Neurodamus,j} + [ATP]_{Metabolism,j} - [ATP]_{j-1} \tag{1}$$

where the subscript $j$ denotes a certain syncing step, in this way we account for the decrement or increment of the ATP quantities from Neurodamus and Metabolism over the full time step $[j-1,j]$. An example of how ATP is sync can be seen in Fig 3B.

- neuronal intracellular ADP at the syncing step $j$ is computed based on the $[ATP]_j$ following the relationship proposed in [14] and present in the original metabolic model [15], as follow:

$$[ADP]_j = \frac{[ATP]_j}{2}\left(-q_{AK} + \sqrt{q_{AK}^2 + 4 \cdot q_{AK} \cdot \left(\frac{A}{[ATP]_j} - 1\right)}\right) \tag{2}$$

where $A \approx 1.44$ mM is the total adenine nucleotide concentration and $q_{AK} = 0.92$ the adenylate kinase equilibrium constant.

**Performance and computational requirements.** A typical simulation of the microcircuit over 3000 ms of simulated time took approximately 5 hours on a HPE SGI 8600 cluster. The Neurodamus framework and the NEURON simulator are optimized for computational efficiency, enabling them to handle the large-scale computations required to simulate microcircuit dynamics. Table 2 summarizes the key parameters and specifications for the simulation.

It should be noted that the simulation's primary bottleneck lies in computational power rather than memory, since the memory footprint is relatively minimal. In fact, the available memory exceeded several terabytes.

**Table 2. Simulation configuration.**

| Parameter | Value |
|---|---|
| CPU Specifications | 2x20 core Intel Xeon Gold Cascade Lake 6248 @ 2.5 GHz |
| Nodes | 64 |
| Tasks per Node | 32 |
| CPUs per Task | 2 |
| MPI Version | HPE MPI (SGI MPT) 2.25 |
| Python Version | 3.8.3 |
| Peak Memory (DDR4 RAM) | 1.71 GB |
| Total Simulation Time (h:m:) | 4h 58m 24s |
| Neurodamus Simulation Time | 2h 16m 1s (46.5%) |
| Metabolism Simulation Time | 1h 42m 42s (35.1%) |
| Syncing, Recording, and Initialization | 53m 42s (18.3%) |

## Supporting information

**S1 Fig. Metabolic model overview.**
(PDF)

**S1 Text. Table A**: m-types used in the circuit and their descriptions. **Table B**: model mechanisms and their respective parameters used for constructing e-models. **Table C**: mechanisms and their neuron model locations for different e-types. **Table D**: parameters of the electrical model used in single neuron models and optimization.
(PDF)

**S2 Text. Tables A–H**: Julia ODEs. **Tables I–X**: Julia Fluxes. **Tables Y–AB**: Initial value of metabolism variables.
(PDF)

**S3 Text. Fig A**: ATP production and consumption per e-type. **Fig B**: Dunn test heatmap corresponding to Fig 4. **Table A**: ATP production and consumption per e-type. **Table B**: Correlation between ATP consumption and production per e-type. **Table C**: Statistical analysis of ATP/s consumption across neural layers. **Table D**: Descriptive statistics across layers for resting membrane potential, spike count, maximum voltage, and action potential (AP) amplitude. **Table E**: Descriptive statistics across e-types for resting membrane potential, spike count, maximum voltage, and action potential (AP) amplitude. **Table F**: Comparison of total spike count and average ATP consumption.
(PDF)

## Acknowledgments

The authors thank Cyrille Favreau and Elvis Boci for their contributions to Fig 1, Weina Ji and Pramod Kumbhar for their support with the software, and Michel Camps and Braeden Benedict for their contributions to the Blue Brain Project. We also extend our thanks to Karin Holm for her editorial assistance.

## Author contributions

**Conceptualization:** Sofia Farina, Polina Shichkova, Henry Markram, Daniel Keller.

**Data curation:** Sofia Farina, Alessandro Cattabiani, Darshan Mandge, Jean Jacquemier.

**Formal analysis:** Sofia Farina.

**Funding acquisition:** Henry Markram.

**Investigation:** Sofia Farina.

**Methodology:** Sofia Farina, Alessandro Cattabiani, Darshan Mandge, Polina Shichkova, James Bryden Isbister.

**Resources:** James G. King, Henry Markram.

**Software:** Alessandro Cattabiani, Polina Shichkova.

**Supervision:** James G. King, Daniel Keller.

**Validation:** Polina Shichkova.

**Visualization:** Sofia Farina, Alessandro Cattabiani.

**Writing – original draft:** Sofia Farina, Alessandro Cattabiani, Darshan Mandge.

**Writing – review & editing:** Polina Shichkova, James Bryden Isbister, Jean Jacquemier, Daniel Keller.

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
