## [Decision Letter · Decision Letter 0]

27 Jan 2025

PCOMPBIOL-D-24-02143

A multiscale electro-metabolic model of a rat neocortical circuit reveals the impact of ageing on central cortical layers

PLOS Computational Biology

Dear Dr. Farina,

Thank you for submitting your manuscript to PLOS Computational Biology. After careful consideration, we feel that it has merit but does not fully meet PLOS Computational Biology's publication criteria as it currently stands. Therefore, we invite you to submit a revised version of the manuscript that addresses the points raised during the review process.

Please submit your revised manuscript within 30 days Mar 29 2025 11:59PM. If you will need more time than this to complete your revisions, please reply to this message or contact the journal office at ploscompbiol@plos.org. Please include the following items when submitting your revised manuscript:

We look forward to receiving your revised manuscript.

Kind regards,

Jonathan Rubin

Academic Editor

PLOS Computational Biology

Hugues Berry

Section Editor

PLOS Computational Biology

**Journal Requirements:**

3) Please ensure that all Table files have corresponding citations and legends within the manuscript. Currently, Table 1 in your submission file inventory does not have an in-text citation. Please include the in-text citation of the table.

Potential Copyright Issues:

i) Figure 1a. Please confirm whether you drew the images / clip-art within the figure panels by hand. If you did not draw the images, please provide (a) a link to the source of the images or icons and their license / terms of use; or (b) written permission from the copyright holder to publish the images or icons under our CC BY 4.0 license. Alternatively, you may replace the images with open source alternatives. See these open source resources you may use to replace images / clip-art:

7) Your current Financial Disclosure states, "The Blue Brain Project, a research center of the École polytechnique fédérale de Lausanne (EPFL), was support by funding from the Swiss government's ETH Board of the Swiss Federal Institutes of Technology (2015-2024). " However, your funding information on the submission form indicates receiving no fund. Please ensure that the funders and grant numbers match between the Financial Disclosure field and the Funding Information tab in your submission form. Note that the funders must be provided in the same order in both places as well.

Please indicate by return email the full and correct funding information for your study and confirm the order in which funding contributions should appear. Please be sure to indicate whether the funders played any role in the study design, data collection and analysis, decision to publish, or preparation of the manuscript.

**Reviewers' comments:**

Reviewer's Responses to Questions

Reviewer #1: The manuscript presents an innovative computational framework combining electrophysiology and metabolism in a reconstructed rat neocortical circuit. The work provides an in-depth analysis of the energy demands of neuronal activity by layers and specific cell types, and provides interesting insights such as how metabolic aging impacts more strongly central cortical layers. This study is particularly noteworthy at the technical level, for its integration of a highly detailed electrophysiological model of cortical circuits with a complex and validated neuro-glia-vascular metabolic model.

While the work is novel and a useful contribution to the field, there are a number of issues that need to be addressed to ensure clarity and validity of the conclusions.

All results appear to rely on two 3-sec simulations, one with constant ATP level and one with the metabolic process. It is therefore unclear if the results are valid for different initial conditions / randomization, for longer durations, and for different states/conditions (e.g. with sensory inputs instead of spontaneous conditions). More than one simulation per condition would be required to quantify results statistically, ensuring they are robust and the metabolic model is not overfitted. Additionally, a longer simulation would demonstrate the results remain valid beyond 3 seconds.

"Layer-specific mitochondrial densities" (Line 109) - Clarify what other layer-specific data was included. Is there any layer-specific data related to the astrocyte morphologies or spatial distribution? This seems relevant given the layer-specific model predictions.

While the results in Fig 5 are interesting, they should be better tied to the rest of the paper; currently, it is not clear why they are relevant and how they relate to the metabolic story.

"Inherent limitations" (line 394) - Please describe these limitations, as this is an important consideration for future model users. Additionally, given several important model limitations previously described in the Discussion, it would be important to clarify which simulation results have been validated against experimental data and which are predictions to be tested.

The Metabolism model in the Methods is not described in enough detail, despite its key role in this study. It would be useful to the reader to see a diagram of the model illustrating the different cellular processes and compartments ("neuronal and astrocytic cytosol, mitochondrial matrix and intermembrane space, interstitium, basal lamina, endothelium, capillaries, arteries, and the endoplasmic reticulum.") The supplementary section with the long list of equations does not help to illustrate the basics of how this model works, so to the reader it remains somewhat as a black box model.

"the energetic demands of the brain account for two-thirds of the body's energy production" (Line 4): Is this correct? The value I commonly see in the literature is 20%.

Fig 1 Panel A is potentially misleading as the vasculature and astrocytes are not modeled spatially in 3D. The caption should make it very clear that this is just a depiction, but the model does not include 3D astrocyte or blood vessel morphologies.

Fig 2: The produced vs consumed colors differ in the legend vs caption.

Fig 6C: The x-axis description is unclear. Is it the number of neurons or spikes?

"NGV unit and solved it for each neuron" (Line 329): Can NGV units be shared among multiple neurons? or is there a 1 to 1 correspondence? Is this realistic?

Reviewer #2: Farina et al. develops a multiscale neuro-glial-vascular model that integrates the electrophysiology of neurons with the metabolic processes. The model is multiscale in time as the fast dynamics of the neuronal electrical activity is coupled with the slower metabolic processes. Another key feature of the model is that the network and morphologies of neuronal models are based on reconstructed rat neocortex. That is, individual neurons are modeled spatially based on their reconstructed morphologies and the network is built on the experimentally observed synaptic densities. The spatial distribution of currents and fluxes follow the observed patterns as well. The model effectively captures electro-metabolic processes at the circuit level, and highlights the importance of heterogeneity within the circuit, where energetic demands vary according to neuronal characteristics. Finally, in metabolic ageing, the model indicates that the middle cortical layers are particularly vulnerable to energy impairment.

Overall, it is an interesting study that provides a valuable platform for investigating neurovascular coupling and the (im)balance between energy demand and supply under different (patho)physiological brain functions. The manuscript and associated code include all details about the model, methods, and results. While normally, I would ask for including more details about the equations used in such complex model and why different equations are chosen, the fact that the authors have provided all equations in Julia format in the manuscript and the code is archived on github sidesteps the concerns about the reproducibility of the results or reusability of the model.

I do not have any major concerns about this manuscript, and recommend the publication of the paper with a few minor edits.

Line 4-5: “Despite its small size, the energetic demands of the brain are remarkably high, accounting for two-thirds of the body’s energy production [1–4].” Please confirm this statement and mention the conditions under which the brain accounts for two-thirds of the energy demand.

Line 86-88: “To address these differences, the two systems were solved independently in an interleaved manner and synchronized every 100 ms.” This is one part of the model that needs more explanation. It is not clear how exactly was the coupling performed at the time step of updating the two systems (tcoupling) and how were the fast variables tracked between the two consecutive larger timesteps at which the coupling was performed (between tcoupling_i and tcoupling_i+1). I am sure it can be figured out from the code, but it’s a key point and researchers might want to use this approach in other models as well. So, explaining this in the manuscript would make it easier.

Line 180: “Synaptic glutamate release pools were also adjusted to 280 account for ageing”. What “adjusted” means? Increased/decreased etc.?

Reviewer #3: The authors propose a novel multiscale model of electro-metabolic coupling in the rat neocortex. Their unique approach includes highly detailed morphologies and various electrical types. The electro-metabolic model is then tested under different scenarios such as resting state, activation and metabolic ageing. The results fully agree with the previous literature, showing that the regions of the brain that are characterised by large synaptic densities are the first to be affected by metabolic ageing. While many simplifying assumptions are used within this model, this work is highly ambitious and could serve as a perfect tool for further development in this novel integrative research area.

Questions and Suggestions:

1. The manuscript is very well written, and every single component of the model is extremely well described. I have found only two typos: legend of Fig 6, panel B (VVisualization) and the second paragraph in the Discussion (Fig.2)-Fig.3).

2. The illustration of the time-coupling is a bit confusing and overly simplified. The time-coupling concept became very clear towards the end of the manuscript, once I read the Multiscale Orchestrator. However, I believe the time-coupling is a core idea in this manuscript and it would be nice for the general readers to understand the concept earlier on. I was wondering whether you could modify fig1C to be more meaningful. I think adding a scheme following the orchestration protocol defined on page 17 would be much more understandable.

3. Throughout the manuscript you repeat that each metabolic step is 100ms and I understood on page 17 that this choice was done by using the Rosenbrock23 solver. Until I reached page 17, I kept wondering how you selected this value. I wonder if you could comment on what would happen if you selected a smaller/larger step. No additional simulations are needed, only a comment.

4. For clarity and fairness, I think you should mention somewhere in the beginning of the article that you only model metabolic ageing and you do not include any morphological changes attributed to ageing. This was not obvious to me until I reached the discussion, and I think it is essential for general readers.

5. It would also be very nice if in the discussion you could provide a hypothesis of how you think changes due to ageing in the morphology could affect your results.

6. On page 4, you mention that the volume fraction of the astrocyte is included in the system of differential equations. Changes in volume fractions during different pathologies such as ischemia and cortical spreading depression have been very well documented in both experimental and computational literature. I was wondering whether you also consider the changes in the extracellular space. Additionally, do these volume fractions also change for the ageing model proposed?

7. I believe not dynamically modelling blood flow is the most significant limitation of your model and might strongly affect your results. Including a blood flow model would increase complexity even further and would add one more timescale to account for. This has actually been done before in https://www.sciencedirect.com/science/article/abs/pii/S0022519319302309 where the authors coupled an electro-metabolic model with a hemodynamic one in which the blood flow is tracked through arteries, capillaries and veins. There is also large literature on hemodynamic models and a lot of experimental data available on rats that could potentially aid you in developing your exciting future work.

**Have the authors made all data and (if applicable) computational code underlying the findings in their manuscript fully available?**

Reviewer #1: Yes

Reviewer #2: Yes

Reviewer #3: Yes

PLOS authors have the option to publish the peer review history of their article (what does this mean?). If published, this will include your full peer review and any attached files.

Reviewer #1: No

Reviewer #2: **Yes: **Ghanim Ullah

Reviewer #3: **Yes: **Gabriela Cirtala

**Figure resubmission:**
---

## [Decision Letter · Decision Letter 1]

19 Apr 2025

Dear Dr. Farina,

We are pleased to inform you that your manuscript 'A multiscale electro-metabolic model of a rat neocortical circuit reveals the impact of ageing on central cortical layers' has been provisionally accepted for publication in PLOS Computational Biology.

Best regards,

Jonathan Rubin

Academic Editor

PLOS Computational Biology

Hugues Berry

Section Editor

PLOS Computational Biology

Reviewer's Responses to Questions

**Comments to the Authors:**

Reviewer #1: Most comments have been succesfully addressed.

Reviewer #3: Thank you for your hard work, improving the text, performing additional simulations and additional figures. I enjoyed reading the improved manuscript and I believe the message and approach are now much clearer. I wish you all the best for the future work.

**Have the authors made all data and (if applicable) computational code underlying the findings in their manuscript fully available?**

Reviewer #1: None

Reviewer #3: Yes

PLOS authors have the option to publish the peer review history of their article (what does this mean?). If published, this will include your full peer review and any attached files.

Reviewer #1: No

Reviewer #3: **Yes: **Gabriela Cirtala

---

## [Editor Report · Acceptance letter]

PCOMPBIOL-D-24-02143R1

A multiscale electro-metabolic model of a rat neocortical circuit reveals the impact of ageing on central cortical layers

Dear Dr Farina,

I am pleased to inform you that your manuscript has been formally accepted for publication in PLOS Computational Biology. Your manuscript is now with our production department and you will be notified of the publication date in due course.

With kind regards,

Lilla Horvath
